# Investigation on Antibiotic-Resistance, Biofilm Formation and Virulence Factors in Multi Drug Resistant and Non Multi Drug Resistant *Staphylococcus pseudintermedius*

**DOI:** 10.3390/microorganisms7120702

**Published:** 2019-12-16

**Authors:** Gabriele Meroni, Joel F. Soares Filipe, Lorenzo Drago, Piera A. Martino

**Affiliations:** 1Department of Veterinary Medicine, Università degli Studi di Milano, 26900 Lodi, Italy; joel.soares@unimi.it (J.F.S.F.); piera.martino@unimi.it (P.A.M.); 2Department of Biomedical Sciences for Health, Università degli Studi di Milano, 20133 Milano, Italy; lorenzo.drago@unimi.it

**Keywords:** antibiotic-resistance, biofilm, *Staphylococcus pseudintermedius*, virulence factors

## Abstract

*Staphylococcus pseudintermedius* is a commensal bacterium frequently isolated from canine skin and recognized as a zoonotic agent especially for dog-owners. This study focused on (a) the antibiotic-resistance phenotypes; (b) the ability to produce biofilm (slime); and (c) the dissemination of virulence factors in *S. pseudintermedius* strains. Seventy-three *S. pseudintermedius* strains were screened for antibiotic-resistance against 22 different molecules by means of Kirby-Bauer assay. The ability to produce biofilm was investigated using the microtiter plate assay (MtP) and the amplification of *icaA* and *icaD* genes. Virulence factors such as cytotoxins (*lukI*), enterotoxins (*seC*), and exfoliative toxins (*siet*, *expA*, and *expB*) were evaluated. The antibiotic-resistance profiles revealed 42/73 (57%) multi-drug resistant (MDR) strains and 31/73 (43%) not-MDR. All the MDR strains and 8/31 (27%) of not-MDR resulted in biofilm producers. Leukotoxin LukI was found in 70/73 (96%) of the isolates. Moreover, the enterotoxin gene *seC* was detected in 47/73 (64%) of the strains. All the isolates carried the *siet* gene, whereas *expA* and *expB* were found in 3/73 (4%) and 5/73 (7%), respectively. In conclusion, *S. pseudintermedius* should be considered a potential zoonotic and human agent able to carry different virulence determinants and capable of producing biofilm which facilitates horizontal gene transfer.

## 1. Introduction

*Staphylococcus pseudintermedius* (SP) is one of the youngest members of the *Staphylococcus* genus, being described and recognized only in 2005 [1], with the first reported case in humans in 2006 [2] and the first molecular identification protocol in 2009 [3]. It is an opportunistic pathogen also known as one of the leading cause of skin, ear, and post-surgical infections in domestic animals, especially in dogs [4,5,6]. Even if infections in humans are less common than those reported in pets, the description of *S. pseudintermedius* as a human pathogen is being increasingly reported [7,8,9], however little is known about its pathogenesis and distribution, and in human medicine is still misdiagnosed as *S. aureus* [10]. A progressive expansion in resistance to commonly prescribed antimicrobial agents has been observed in the past years, in particular with the emergence and the global spread of multi drug resistant (MDR) bacteria with particular regards for the Methicillin Resistant *S. pseudintermedius* (MRSP) clones that dramatically complicate the treatment of these infections [7,11].

The ability to form biofilm is one of the major virulence determinants studied nowadays in bacteria because it facilitates the adherence to biotic and/or abiotic surfaces [12]. Biofilm-related infections are fastidiously faced because sessile bacteria are generally much more tolerant to antibiotics compared to the equivalent planktonic forms and can easily resist to host immune responses [13]. To form a biofilm, bacteria require at least two features: a) Adherence to a surface, and b) the possibility to accumulate it in order to form a multi-layered complex structure. The ability to form a biofilm resides in the formation of an extracellular matrix known as polysaccharide intercellular adhesion molecule (PIA), that is encoded by the *ica* operon, including four distinct genes (A, B, C, and D) [7,12,14]. The biofilm forming ability of *S. pseudintermedius* has been reported by different authors but nowadays is still not fully understood; in a recent study Stefanetti et al [11,15], up to 96% of canine SP strains was able to produce biofilm. Moreover, *ica*-independent biofilm formation has been reported in staphylococci [16,17,18].

The aim of the present study is to analyze the possible correlation between antibiotic-resistance, ability to form biofilm, and dissemination of virulence determinant in different isolated strains of SP.

## 2. Materials and Methods

### 2.1. Bacterial Isolation and Identification

The 73 *S. pseudintermedius* strains included in this study were collected between 2016 and 2018 at the Microbiology Laboratory of the Department of Veterinary Medicine (Università degli Studi di Milano). Clinical samples were cultivated on Trypticase Soy Agar (TSA) +5% defibrinated sheep blood agar (Microbiol, Uta, Sardinia, CA, Italy) and incubated aerobically at 37 °C for 24 h. Following morphological analysis, the suspected staphylococcal colonies were sub-cultured on Mannitol Salt Agar (MSA; Microbiol, Uta, Sardinia, CA, Italy) for genus identification and incubated at 37 °C for 24 h. To confirm the isolation of *S. pseudintermedius*, standard phenotypic techniques were used such as Gram stain, catalase test, and coagulase test. Finally, the amplification of the *nuc* gene (Table 1) described by Sasaki in 2010, was used to genetically confirm the isolates at species level [19]. All the strains were stored at −20 °C in 25% glycerol.

### 2.2. DNA Extraction

Pure cultures stocks in glycerol were thawed at room temperature and grown on blood agar plates at 37 °C for 24 h. A single colony was picked up and grown in Brain Heart Infusion Broth (BHI, Scharlau, Spain) at 37 °C for 8 h. One mL aliquot was used for DNA extraction using the previously described boiling method [20]. DNA was quantified and checked for its purity using the NanoDropTM 2000 Spectrophotometer (Thermo Fisher Scientific, Monza (MB), Lombardy, Italy).

### 2.3. Molecular Typing

In order to better characterize the *S. pseudintermedius* strains, two commonly used typing techniques were used: MultiLocus Sequence Typing (MLST) and SCCmec Typing.

Genetic diversity of the strains was determined by MLST of seven genes (*tuf*, *cpn60*, *pta*, *purA*, *fdh*, *ack*, *sar*); the primers used as well as the amplification conditions were the same as those previously described by Solyman et al. 2013 [21]. MLST sequences were aligned with sequences present in the NCBI nucleotide database in order to set out the allele number. Sequence types (STs) were assigned according to the literature [22] and using *Staphylococcus pseudintermedius* MLST database (https://pubmlst.org/spseudintermedius/).

SCCmec types I–VI were assigned, among MRSP strains only, using a specific set of multiplex PCR assays as reported by different authors, using the same set of primers and amplification conditions previously described [23,24].

### 2.4. Determination of Antibiotic-Resistance Profile

#### 2.4.1. Kirby-Bauer Disk Diffusion Method

Susceptibility to a panel of 22 antimicrobial agents was determined by the Kirby-Bauer disk diffusion test according to the Guidelines of the Clinical Laboratory and Standards Institute (CLSI, 2015). Disks of 22 different antibiotics were used as reported below (in brackets the concentration in µg): Oxacillin (OX, 5), Amoxicillin + Clavulanic acid (AMX, 30 [20/10]), Amoxicillin (AML, 30), Carbenicillin (CAR, 100), Cephalexin (CL, 30), Cefovecin (CVN, 30), Ceftiofur (EFT, 30), Ceftriaxone (CRO, 30), Clindamycin (DA, 10), Lincomycin + Spectinomycin (MY, 15 [5/10]), Doxycycline (DO, 5), Enrofloxacin (ENR, 5), Marbofloxacin (MAR, 5), Pradofloxacin (5, 5), Amikacin (AK, 30), Gentamicin (CN, 30), Neomycin (N, 30), Tobramycin (TOB, 10), Kanamycin (K, 30), Rifampicin (RD, 30), Azithromycin (AZM, 15), Erythromycin (E, 30). The results were recorded as susceptible, intermediate, or resistant by the measurement of the inhibition halo diameter.

#### 2.4.2. Amplification of Antibiotic-Resistance Genes (ARg)

The primers used for the detection of five different antibiotic-resistance genes (ARg) were taken from the literature [25,26,33], synthesized by Eurofins Genomics, and listed in Table 1. Predicted amplicon size and primers specificity were defined using BLAST search available through the National Center for Biotechnology Information website (www.ncbi.nlm.nih.gov) and coupled with BioEdit freeware software. To check specificity on DNA from genotypically defined isolates, single PCRs for each primer pair were performed before the Multiplex PCR assay.

For *tetK, tetM,* and *aacA-aphD* genes, the protocol described by Strommenger in 2003 was followed [26]. Multiplex PCR amplifications were carried out with AccuPrime^TM^ Taq DNA Polymerase system (Invitrogen, Italy) following the manufacturer′s instruction in a 25 µL volume comprising approximately 40 ng of DNA, 10 pmol of each of 6 primers, 2.5 µL of 10× AccuPrime^TM^ PCR Buffer II, 4 nM of MgCl_2_ (final concentration), 0.3 U of AccuPrime^TM^ Taq DNA Polymerase, and nuclease-free water (NFW) to reach the final volume.

For *mecA* and *blaZ* genes, the protocol used was described by Kang in 2014 [25]. Multiplex amplifications were carried out using the same kit previously described. The PCR products were resolved on a 1.5% agarose gel (GellyPhore^LE^, EuroClone, Italy).

### 2.5. Biofilm Analysis

#### 2.5.1. Identification of Biofilm-Forming Strains

The identification of biofilm-forming strains was carried out by the microtiter plate (MtP) assay, as previously described [34,35]. Briefly, after growing in BHI broth at 37 °C for 24 h, pure staphylococcal cultures were 1:100 diluted in fresh Trypticase broth (Oxoid, Italy) + 1% glucose (TSBg) and seeded in 96 well-plates (Corning, USA). After 24 h of incubation at 37 °C, planktonic bacteria were washed out and biofilm was stained with Crystal violet (Carlo Erba, Italy). Negative controls consist of TSBg only. Each strain was analyzed in triplicate on the same plate and three independent plates were used. The absorbance (570 nm) of negative controls was used to set the optical density cut-off (ODc) as three standard deviations above the mean OD of the negative control. Strains were classified as follows: Not adherent OD ≤ ODc; weakly adherent ODc < OD ≤ 2 × ODc; moderately adherent ODc < OD ≤ 4 × ODc; strongly adherent OD > 4 × ODc.

#### 2.5.2. Amplification of Biofilm-Associated Genes and *Agr*-Typing

To confirm data from MtP assay, the detection of two pivotal genes of *ica locus* (*icaA* and *icaD*) was performed by conventional qualitative PCR (see Table 1) using primer according to literature [14]. Taking into account that the ability to produce biofilm could be associated with a specific antibiotic-resistance profile [7,36], the *agr locus* was analyzed with two duplex PCRs for the determination of *agr* type (I-IV), as previously reported [27]. For the two duplex PCRs, the already described AccuPrime^TM^ Taq DNA Polymerase system (Thermo Fisher Scientific, Monza (MB), Lombardy, Italy) was used.

### 2.6. Virulence Factors Carriage

To establish the pathogenicity of the isolated strains, specific virulence factors (*lukS-F*, *seC*, *siet*, *expA* and *expB*) were searched by qualitative PCR. Their specific thermal cycling conditions and primer pairs are listed in Table 1.

### 2.7. Statistical Analysis

Statistical significance was determined by GraphPad Prism v6 (GraphPad Software^®^, La Jolla CA, USA) using Fisher′s exact test. Data were analyzed by contingency tables (2-by-2 layout). A *p-value* < 0.05 was considered significant. The agreement between the MtP assay and PCR detection of *icaA* and *icaD* genes was calculated using Cohen’s Kappa values for dichotomous data in Microsoft Excel. The strength of the accordance was interpreted according to Landis and Koch (1977), who classified agreement in the following categories: 0–0.2 as poor; 0.21–0.4 as fair; 0.41–0.6 as moderate, 0.61–0.8 as good; 0.81–1 as very good [37].

## 3. Results

### 3.1. Molecular Identification of S. pseudintermedius, MLST, and SCCmec Typing

All the isolates were from dogs with clinical deep pyoderma, a single 926 bp fragment was derived from the amplification of themonuclease gene (*nuc*) confirming, at species level, the phenotypic isolation of 73 *S. pseudintermedius* strains.

The 73 isolates were assigned to three different STs with the following prevalences: ST 71 56/73 (77%), ST 258 12/73 (16.4%), and ST 106 5/73 (6.6%).

SCCmec types were assigned to 35/76 (48%) strains that resulted positive to *mecA* gene detection and classified as methicillin-resistant *S. pseudintermedius* (MRSP), resulting in two different types, as reported in Table 2.

### 3.2. Overall Antibiotic-Resistance

The antibiotic-resistance of all 73 *S. pseudintermedius* isolates is shown in Figure 1. Kirby-Bauer assay demonstrated that 42/73 (57.5%) of SP isolates were MDR, exhibiting resistance against macrolides (97%), fluoroquinolones (86%), and β-lactams (72%). The most in-vitro effective molecules were amikacin (100% susceptible), rifampicin (93% susceptible), and partially gentamicin (45% susceptible). The remaining 31 not-MDR strains had a low prevalence of resistance for all the antibiotics with the exception of clindamycin which had a rate of resistance of 35% (11/31). Table 3 shows the prevalence of ARg by using Multiplex PCRs.

### 3.3. Biofilm Formation Assay

Figure 2A shows the difference in terms of number of non-biofilm producing strains among not-MDR and MDR strains. Between these two groups, there is a clear difference in terms of the ability to produce biofilm (Figure 2B). The MDR bacteria were all able to produce exopolysaccharide, resulting in 20/43 (46.5%) strong biofilm producers, 20/43 (46.5%) moderate, and 2/43 (4.6%) weak producers. In the not-MDR group, only 1/31 (3.22%) strains was strong producer, 9/31 (29%) strains were categorized as moderate producers, and 17/31 (55%) resulted in weak biofilm producers. The remaining 4/31 (13%) did not show slime production. At the molecular level, the presence of *icaA* and *icaD* genes was demonstrated by the amplification of the corresponding amplicons. Both the *icaA* and *icaD* genes were detected in 30/31 (97%) not-MDR strains, while 28/42 (66.6%) of MDR bacteria had both the targeted genes. Among the remaining 14 strains, one was negative for both these genes (but still able to produce biofilm) and the remaining 13 were all positive for *icaD* only. Neither one of these genes was detected in 2/73 (2.7%) isolates (details are reported in Appendix A). The agreement between the microplate assay and the amplification of *ica locus* genes was not significant (k < 0.01). The determination of *agr*-typing showed that all the 73 SP strains belong to the *agr* type *I*.

### 3.4. Virulence Factors

Table 2 shows the distribution of all the genes analyzed in this study. Among the virulence factors, *Chi*-squared test showed differences in *icaA* and *seC* genes between MDR and not-MDR bacteria, for all the other virulence determinants, no differences were found (Figure 3).

## 4. Discussion

Antibiotic-resistance remains one of the most important problems to face within treatment and control of *S. pseudintermedius* related infection in human and in veterinary medicine. SP is nowadays considered a potential zoonotic agent able to colonize also humans and is regarded as one of the increasing skin and soft tissue-related pathogens [9,38,39]. MDR strains are defined when “resistant to at least 1 agent in 3 or more antimicrobial categories” [40]. In this study, 57.5% (42/73) of the strains had MDR phenotype, also confirmed by the detection of AR genes. The majority of the resistant strains 35/42 (83%) resulted positive for *mecA* gene, associated with oxacillin resistance; this result is in concordance with a previous study which revealed a 100% of *mecA* gene amplification in MRSP strains [41]. A large portion of isolates showed a pattern of resistance against β-lactams, macrolides, and fluoroquinolones.

A total of three STs (ST 71, ST 258 and ST 106) were detected among the 73 SP strains. This result is not totally concordant with the literature in which different authors found a higher genetic diversity in the analyzed population (in terms of STs assigned) [42,43]. ST 71 is predominant and was assigned to 56/73 (77%) strains; this particular sequence type remains the most abundant in Europe, while ST 68 is predominant in United States, and ST 45/ST 112 in Asia [21,22,23,24,25,26,27,28,29,30,31,32,33,34,35,36,37,38,39,40,41,42,43,44]. In Europe, between 2012–2013 and 2015–2016, the prevalence of ST 71 rapidly decreased from 65.3% to 55.2%, whereas the emergence of ST 258, originally derived from Northern Europe, was described (from 1.1% to 5.78%) [43]. ST 71 was highly resistant to antibiotics, showing multiple resistances against commonly veterinary-licensed antibiotics (e.g., tetracyclines). On the other hand, ST 258 was reported to be more frequently susceptible to antibiotics (e.g., enrofloxacin, gentamicin) [43,44].

In this study, all the MRSP strains (35/73; 48%) resulted in MDR, and the majority (68.5%) were classified as ST 71. SCCmec typing showed two chromosomal cassettes types: II-III present in all ST 71, and IV detected in the other MRSP strains belonging to STs 258/106 (Table 2).

A clear correlation was found between antibiotic-resistance and the ability to produce biofilm, suggesting that MDR staphylococci are more prone to produce large quantities of slime. This particular result is in accordance with other studies and could be reasonably explained by the presence of transposons which are the mobile element that easily can be exchanged between strains [11,12,14].

The majority (94.5%) of the 73 isolates were biofilm producers, and this result is in agreement with the current literature on biofilm in *Staphylococcus* clinical isolates [14,45]. Moreover, a genetic approach revealed the concomitant presence of either *icaA* or *icaD* in 79.5% (58/73) of the strains studied, while 18% (13/73) of the isolates presented the amplification of *icaD* and only two strains were negative for both these two genes but still able to produce biofilm. Authors suggest that the presence of *icaD* only (found in 42/43 MDR strains) is sufficient to produce exopolysaccharide, as demonstrated for *S. epidermidis* [12]. These findings are discordant with a previous study [14] in which the author suggested that biofilm formation occurred only when both *icaA* and *icaD* are expressed, but however concordant with another study [15] which reported the absence of correlation between the presence of both *ica* genes and biofilm formation (*ica*-independent biofilm producers strains). These results suggest the importance of combined, phenotypic, and genetic methods for checking biofilm formation in *S. pseudintermedius*.

The prevalence of *agr* groups in our clinical isolates is very different to that described by Little et al. [7], with *agrI* being the unique group found in our collection, while any of the strains were assigned to one of the other three (*agr II*, *III* and *IV*). Following the suggestions of the author, in this contest, it is not possible to correlate the ability to produce biofilm with the corresponding *agr* group. Little et al. (2019) stated that strains harboring *agrI* were more prone to produce biofilm and to be MDR [7]. Our findings indicate a major ability for MDR bacteria to be strong biofilm producers.

Among the virulence determinants analyzed in this study, only the prevalence of specific enterotoxin *seC* was found statistically significant between MDR and not-MDR strains (*p-value*: 0.016). The prevalence of individual genes was similar to the literature [28,29].

## Figures and Tables

**Figure 1 microorganisms-07-00702-f001:**
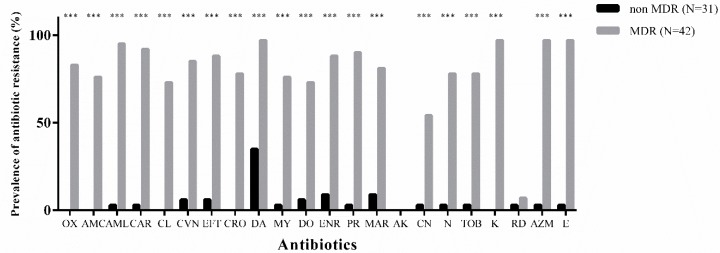
Overall prevalence of antibiotic-resistance among multi-drug resistant (MDR) and not-MDR isolates. The *Chi*-squared test showed statistical differences among the two groups for the majority of antibiotic molecules tested except for amikacin and rifampicin. (***: *p-value* < 0.001).

**Figure 2 microorganisms-07-00702-f002:**
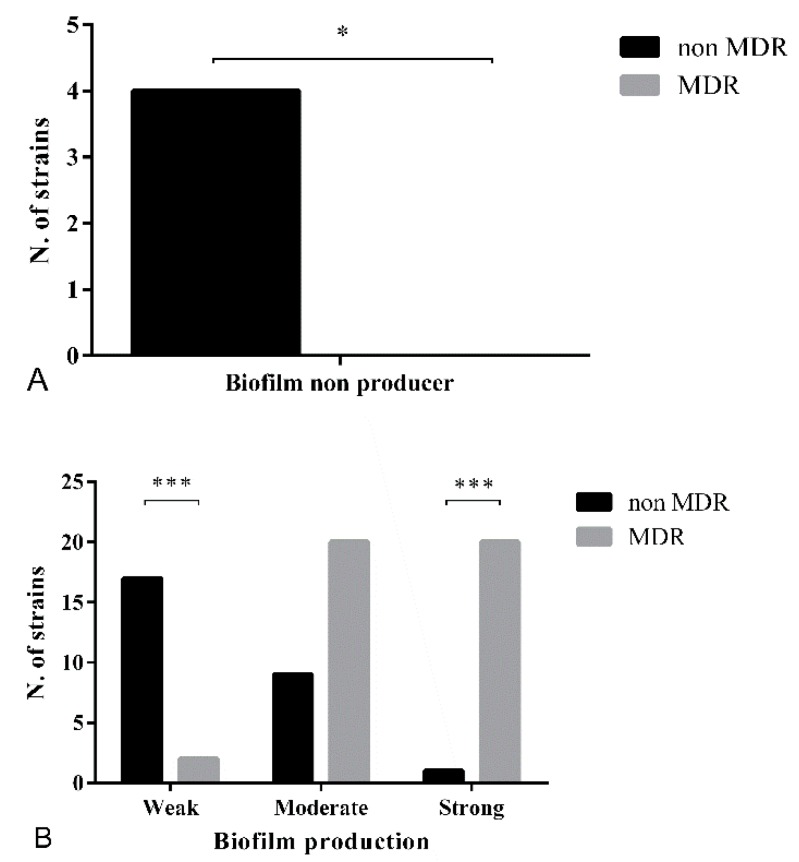
Biofilm forming ability in not-MDR and MDR *Staphylococcus*
*pseudintermedius* (SP) strains. (**A**) All the MDR strains are able to produce biofilm and these data are statistically significant compared to the not MDR group in which 4 strains resulted in non-biofilm producers. (**B**) The majority of MDR strains were strongly biofilm producers (20/42), whereas not-MDR strains were mostly categorized as weak slime producers. (*: *p-value* between 0.05 and 0.01; ***: *p-value* < 0.001).

**Figure 3 microorganisms-07-00702-f003:**
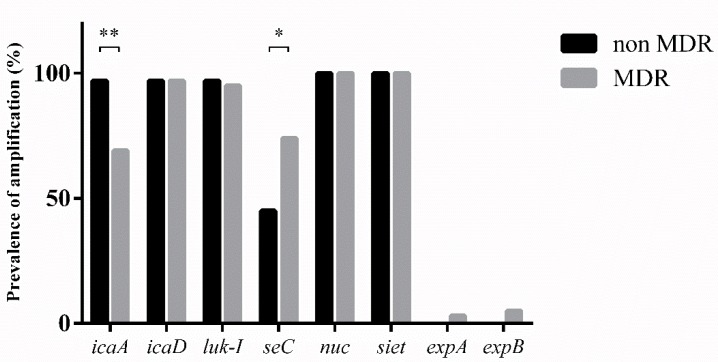
Dissemination of virulence factors among MDR and not-MDR strains. (*: *p*-value between 0.05 and 0.01; **: *p-value* < 0.01).

**Table 1 microorganisms-07-00702-t001:** Primers, amplicon size, and amplification conditions.

	Genes		Sequence (5’-3’)	Amplicon Size (bp)	PCR Conditions	References
Antibiotic-resistance genes	*mecA*	F	AAAATCGATGGTAAAGGTTGGC	532	95 °C × 4 min, 30 × (95 °C × 1 min, 58 °C × 1 min, 72 °C × 1 min) 72 °C × 7 min, 4 °C	[25]
		R	AGTTCTGCAGTACCGGATTTGC			
	*blaZ*	F	TGACCACTTTTATCAGCAACC	750		
		R	GCCATTTCAACACCTTCTTTC			
	*tetK*	F	GTAGCGACAATAGGTAATAGT	360	94 °C × 3 min, 30 × (94 °C × 30 s, 55 °C × 30 s, 72 °C × 30 s) 72 °C × 7 min, 4 °C	[26]
		R	GTAGTGACAATAAACCTCCTA			
	*tetM*	F	AGTGGAGCGATTACAGAA	158		
		R	CATATGTCCTGGCGTGTCTA			
	*aacA-aphD*	F	TAATCCAAGAGCAATAAGGGC	227		
		R	GCCACACTATCATAACCACTA			
Biofilm genes	*icaA*	F	ACTGTTTCGGGGACAAGCAT	134	94 °C × 3 min, 35 × (94 °C × 15 s, 60 °C × 20 s, 72 °C × 20 s) 72 °C × 7 min, 4 °C	[14]
		R	ATTGAGGCTGTAGGGCGTTG			
	*icaD*	F	CGTTAATGCCTTCTTTCTTATTGCG	166	94 °C × 3 min, 35 × (94 °C × 15 s, 56 °C × 20 s, 72 °C × 20 s) 72 °C × 7 min, 4 °C	
		R	ATTAGCGCACATTCGGTGTT			
*Quorum-sensing* genes	*pan-agr*	F	ATGCACATGGTGCACATGC		94 °C × 3 min, 35 × (94 °C × 15 s, 56 °C × 20 s, 72 °C × 20 s) 72 °C × 7 min, 4 °C	[27]
	*agrI*	R	GTCACAAGTACTATAAGCTGCGAT			
	*agrII*	R	GTATTACTAATTGAAAAGTGCCATAGC			
	*agrIII*	R	CTGTTGAAAAAGTCAACTAAAAGCTC			
	*agrIV*	R	CGATAATGCCGTAATACCCG			
Virulence factors	*luk-F*	F	CCTGTCTATGCCGCTAATCCA	572	94 °C × 3 min, 35 × (94 °C × 1 min, 57 °C × 1 min, 72 °C × 1 min) 72 °C × 7 min, 4 °C	[28]
		R	AGGTCATGGAAGCTATCTCGA			
	*luk-S*	F	TGTAAGCAGCAGAAAATGGGG	503		
		R	GCCCGATAGGACTTCTTACAA			
	*seC*	F	GGCGGCAATATTGGCGCTCG	271	95 °C × 2 min, 30 × (95 °C × 1 min, 55 °C × 1 min, 72 °C × 2 min) 72 °C × 5 min, 4 °C	[29]
		R	TTACTGTCAATGCTCTGACC			
	*nuc*	F	TRGGCAGTAGGATTCGTTAA	926	95 °C × 2 min, 30 × (95 °C × 30 s, 52 °C × 30 s, 72 °C × 30 s) 72 °C × 2 min, 4 °C	[19]
		R	CTTTTGTGCTYCMTTTTGG			
	*siet*	F	ATGGAAAATTTAGCGGCATCTGG	359	94 °C × 3 min, 30 × (94 °C × 30 s, 56 °C × 30 s, 72 °C × 1 min) 72 °C × 5 min, 4 °C	[30]
		R	CCATTACTTTTCGCTTGTTGTGC			
	*expA*	F	GTKTTAATTGGWAAAAATACA	413	94 °C × 3 min, 30 × (94 °C × 1 min, 42 °C × 1 min, 72 °C × 1 min) 72 °C × 4 min, 4°C	[31]
		R	ATNCCWGAKCCTGAATTWCC			
	*expB*	F	GGGCATGCACATATGATGAAGCC	820	95 °C × 3 min, 30 × (95 °C × 1 min, 53 °C × 1 min, 72 °C × 1 min) 72 °C × 4 min, 4 °C	[32]
		R	CCAGATCTATCTTCTGATTCAGC			

**Table 2 microorganisms-07-00702-t002:** Molecular characterization of the 35 methicillin-resistant *S. pseudintermedius* (MRSP) isolates.

MLST	SCCmec Types	No. of Isolates (%)
ST 71	II-III	24 (68.5%)
ST 258	IV	9 (25.7%)
ST 106	IV	2 (5.7%)

**Table 3 microorganisms-07-00702-t003:** Distribution of antibiotic-resistance and virulence factors genes.

	Genes	not MDR Strains	MDR Strains	*p-Value*
Antibiotic-resistance genes	*mecA*	0	35/42 (83%)	**<0.0001**
	*blaZ*	7/31 (23%)	42/42 (100%)	**<0.0001**
	*tetK*	2/31 (6.4%)	13/42 (31%)	**0.0171**
	*tetM*	0	22/42 (50%)	**<0.0001**
	*aacA-aphD*	5/31 (16.6%)	32/42 (76%)	**<0.0001**
Biofilm genes	*icaA*	30/31 (97%)	29/42 (69%)	**0.0026**
	*icaD*	30/31 (97%)	41/42 (97%)	>0.05
Virulence factors	*luk-I*	30/31 (97%)	40/42 (95%)	>0.05
	*seC*	14/31 (45%)	31/42 (74%)	**0.016**
	*nuc*	31/31 (100%)	42/42 (100%)	>0.05
	*siet*	31/31 (100%)	42/42 (100%)	>0.05
	*expA*	0	3/42 (7%)	>0.05
	*expB*	0	5/42 (12%)	>0.05

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
