# Peer review of "Investigation on Antibiotic-Resistance, Biofilm Formation and Virulence Factors in Multi Drug Resistant and Non Multi Drug Resistant Staphylococcus pseudintermedius"

_microorganisms, 2019, doi:10.3390/microorganisms7120702_

Round 1

Reviewer 1 Report

1) Some of the references in the introduction seem very old, eg: "Futagawa-Saito K, Sugiyama T, Karube S, Sakurai N, Ba-Thein W, Fukuyasu T. Prevalence and 196 characterization of leukotoxin-producing Staphylococcus intermedius in isolates from dogs and pigeons. J 197 Clin Microbiol. 2004;42(11):5324–6." Can authors update to last 10 years of references.

2) Is there any specific reason to consider Cohen’s Kappa values for detecting agreement between the MtP assay and PCR detection? If so, please mention them in methods.

3) Why only two specific virulence factors lukS-F 50 and seC were selected for determining pathogenicity of the isolated strains?

4) Is there any specific reason to consider only 5 QS genes? Authors can consider more to validate the hypothesis. This seems a selection bias?

5) Two biofilm genes does not have significant correlation with antibiotic resistance, did authors tried to explore them and study the reasons and mechanisms?

Author Response

Dear reviewer,

Below you can read our answers/comments:

1) Some of the references in the introduction seem very old, eg: "Futagawa-Saito K, Sugiyama T, Karube S, Sakurai N, Ba-Thein W, Fukuyasu T. Prevalence and 196 characterization of leukotoxin-producing Staphylococcus intermedius in isolates from dogs and pigeons. J 197 Clin Microbiol. 2004;42(11):5324–6." Can authors update to last 10 years of references.

We'll revise the introduction and when possible change the references with newer ones (eg. Maali et al., Understanding the virulence of S. pseudintermedius: A major role of pore-forming toxins (2018).)

2) Is there any specific reason to consider Cohen’s Kappa values for detecting agreement between the MtP assay and PCR detection? If so, please mention them in methods.

Cohen's Kappa test was used to compare the result of both assays, this was done based on the fact that this test is used as "a quantative measure of agreement between two methods with binary outcomes" Cohen, 1960 (as in our case). This is the same approach used by Casagrande Proietti et al. 2015.

3) Why only two specific virulence factors lukS-F 50 and seC were selected for determining pathogenicity of the isolated strains?

We studied other virulence factors (SIET, ExpA, and ExpB) but with really low prevalences; those will be included in the next paper version.

4) Is there any specific reason to consider only 5 QS genes? Authors can consider more to validate the hypothesis. This seems a selection bias?

Those primers were selected only to type agr locus (agrABCD) (they are actually 1 forward and 4 reverse primers, designed between agrC and agrD genes). The amplification classify the strain in 4 agr types. We didn't amplify the two others agr genes (A and B).

5) Two biofilm genes does not have significant correlation with antibiotic resistance, did authors tried to explore them and study the reasons and mechanisms?

No.

Reviewer 2 Report

The paper described some characteristics of strains of Staphylococcus pseuintermedius isolated recently in Milan. It reinforces the well established knowledge that this organism is often resistant to multiple antibiotics including methicillin (oxacillin, all beta lactams). There is a growing interest in this organism as a cause of zoonotic infections. Unfortunately this study does little to advance knowledge for several reasons  

1.The clinical background of the isolates was not reported. 

2. State of the art molecular typing was not performed. The strains should have been classified genetically by multilocus sequence typing and the STs reported. Then the genertic diversity of the collection would been known and correlations with resistance, virulence factors and agr type could have been made

3. The full repertoire of virulence factors was not reported.  Indeed use of PCR for molecular typing is problematic because of false negatives and lack of discrimination. Nowadays whole genome sequencing is the gold standard for detailed characterization of clinical isolates. 

Author Response

Dear Reviewer,

Below you can find our answers/comments:

1.The clinical background of the isolates was not reported.

All of ours samples are from canine cutaneous swabs from pyoderma cases, however no other clinical history information has been given.

2. State of the art molecular typing was not performed. The strains should have been classified genetically by multilocus sequence typing and the STs reported. Then the genertic diversity of the collection would been known and correlations with resistance, virulence factors and agr type could have been made

MLST and SCCmec typing were performed and we'll provide the results in a new specific section of the manuscript.

Unfortunately, we cannot correlate the molecular profiles with resistance, virulence factors and agr type, because we don't have a correspondance between each strain and their relative molecular profile. The goal of our research didn't include genetic population study, as you correctly suggested. 

3. The full repertoire of virulence factors was not reported. Indeed use of PCR for molecular typing is problematic because of false negatives and lack of discrimination. Nowadays whole genome sequencing is the gold standard for detailed characterization of clinical isolates.

For economic and logistic reasons, it was not possible to perform NGS analysis. However, comparing our results with literature we're confident about our data.

Reviewer 3 Report

General Considerations: In the manuscript “Investigation on antibiotic-resistance, biofilm formation and virulence factors in Multi Drug Resistant and non Multi Drug Resistant Staphylococcus pseudintermedius” the authors performed a phenotypic and molecular characterization of 73 clinical isolates of the opportunistic pathogen S. pseudintermedius. Considering the increasing prevalence of antibiotic resistant strains and the pathogenic potential of S. pseudintermedius for pets, namely dogs, several studies have been carried out worldwide to investigate the antibiotic resistance and biofilm formation of these species. This kind of studies is usually important to limit the use of some antimicrobials and to adapt the veterinary practice. The manuscript is objective, and it is generally well written and clearly presented. However, in the discussion the importance of this study should be mentioned, and the significance of your findings should be explored.

Comments:

1.       Problems with page numbering due to landscape page.

2.       Use dots and not commas in decimals.

3.       Section 2.4, Line 31: Instead of “The identification of exopolysaccharide-forming strains” à “The identification of biofilm-forming strains”

4.       Figure 1: Please check if there are no statistical differences for Kanamycin. 

5.       Section 3.2, Line 75: “Exopolysaccharide formation assay” à “Biofilm formation assay”

6.       Section 3.2, Line 78: “The MDR bacteria were all able to produce exopolysaccharide”. Could you evaluate the exopolysaccharide production using the microtiter plate (MtP) assay? The ability of bacteria to aggregate and form biofilm is strictly related to the capacity of producing an extracellular mucoid substance, but just use the microtiter plate assay to state this can be exploitative.

7.       Section 3.2, Line 83 and 84: “icaA and icaD genes were both detected in all not-MDR strains”. In Supplementary Table S1 there is 1 strain (ID 31) that is negative for both genes.

8.       Section 3.2, Line 85: “Among the remaining 13 strains, 1 was negative for both these genes (but still able to produce biofilm) and the remaining 12 were all positive for icaD only.” Actually, in Supplementary Table S1 there are 14 remaining strains, 13 of them are positive for icaD only.

9.       As mentioned, recently Little and co-authors analyzed the prevalence and distribution of agr groups for 160 S. pseudintermedius canine clinical isolates (100% identity to 1 of the 4 known agrD alleles). In order to determine the agr group, PATRIC’s BLASTn function was used on the assembled genomes using previously published sequences for the 4 agrD alleles (GenBank accession no. EU157356.1, EU157391.1 , EU157400.1, and EU157402.1) (as described by Bannoehr et al., 2007). In your study, the agr specific groups were identified using a strategy designed for S. aureus. Could the selected strategy be responsible for the differences (misidentification) of the agr groups in your clinical isolates?  

Little SV, Bryan LK, Hillhouse AE, Cohen ND, Lawhon SD. 2019. Characterization of agr groups of Staphylococcus pseudintermedius isolates from dogs in Texas. mSphere 4:e00033-19. https://doi.org/10.1128/ mSphere.00033-19

Bannoehr J, Ben Zakour NL, Waller AS, Guardabassi L, Thoday KL, van den Broek AH, Fitzgerald JR. 2007. Population genetic structure of the Staphylococcus intermedius group: insights into agr diversification and the emergence of methicillin-resistant strains. J Bacteriol 189: 8685– 8692. https://doi.org/10.1128/JB.01150-07.

Author Response

Dear Reviewer,

Below you can find our answers/comments:

1.Problems with page numbering due to landscape page.

We appreciate your comment about the page numbering problem, and in the final version we'll modify it.

2. Use dots and not commas in decimals.

In the new version we'll standardize and use only dots for decimals-

3. Section 2.4, Line 31: Instead of “The identification of exopolysaccharide-forming strains” à “The identification of biofilm-forming strains”

We think you're talking about line 113 and we'll change it according with your suggestion

4. Figure 1: Please check if there are no statistical differences for Kanamycin.

An error during the figure preparation was made, and there is a statistical difference for Kanamycen but no statistical difference was observed for Rifampicin (RD). We're also providing changes in the test where those antibiotics have been mentioned.

5. Section 3.2, Line 75: “Exopolysaccharide formation assay” à “Biofilm formation assay”

As for comment 3, we'll change it according with your suggestion

6. Section 3.2, Line 78: “The MDR bacteria were all able to produce exopolysaccharide”. Could you evaluate the exopolysaccharide production using the microtiter plate (MtP) assay? The ability of bacteria to aggregate and form biofilm is strictly related to the capacity of producing an extracellular mucoid substance, but just use the microtiter plate assay to state this can be exploitative.

Yes, we evaluated the biofilm forming ability with the MtP assay, and coupled it with the amplification of icaA and icaD responsible for exopolysaccharide prodution.

7. Section 3.2, Line 83 and 84: “icaA and icaD genes were both detected in all not-MDR strains”. In Supplementary Table S1 there is 1 strain (ID 31) that is negative for both genes.

You're correct, ID31 is negative for both genes, and this is one of the cases of ica independent biofilm production, as reported in our discussion

8. Section 3.2, Line 85: “Among the remaining 13 strains, 1 was negative for both these genes (but still able to produce biofilm) and the remaining 12 were all positive for icaD only.” Actually, in Supplementary Table S1 there are 14 remaining strains, 13 of them are positive for icaD only.

The phrase will be replaced with the correct numbers "Among the remaining 14 strains, 1 was negative for both these genes (but still able to produce biofilm) and the remaining 13 were all positive for icaD only."

9. As mentioned, recently Little and co-authors analyzed the prevalence and distribution of agr groups for 160 S. pseudintermedius canine clinical isolates (100% identity to 1 of the 4 known agrD alleles). In order to determine the agr group, PATRIC’s BLASTn function was used on the assembled genomes using previously published sequences for the 4 agrD alleles (GenBank accession no. EU157356.1, EU157391.1 , EU157400.1, and EU157402.1) (as described by Bannoehr et al., 2007). In your study, the agr specific groups were identified using a strategy designed for S. aureus. Could the selected strategy be responsible for the differences (misidentification) of the agr groups in your clinical isolates?

Even if we used primers designed for S. aureus, before the use of those primers in our lab we confirmed that they were also specific for S. pseudintermedius (this locus is highly conserved among Staphylococci) and we aligned the primers with the S. pseudintermedius genome (BLAST).

Round 2

Reviewer 1 Report

Authors seem to have addressed the concerns I have raised and I accept these changes. I however look forward to authors new article with tested other factors as mentioned: "We studied other virulence factors (SIET, ExpA, and ExpB) but with really low prevalences; those will be included in the next paper version."

Reviewer 2 Report

No further comments